# Creation of energetic biothermite inks using ferritin liquid protein

Joseph M. Slocik[1], Ruel McKenzie[1], Patrick B. Dennis[1] & Rajesh R. Naik[2]

Energetic liquids function mainly as fuels due to low energy densities and slow combustion kinetics. Consequently, these properties can be significantly increased through the addition of metal nanomaterials such as aluminium. Unfortunately, nanoparticle additives are restricted to low mass fractions in liquids because of increased viscosities and severe particle agglomeration. Nanoscale protein ionic liquids represent multifunctional solvent systems that are well suited to overcoming low mass fractions of nanoparticles, producing stable nanoparticle dispersions and simultaneously offering a source of oxidizing agents for combustion of reactive nanomaterials. Here, we use iron oxide-loaded ferritin proteins to create a stable and highly energetic liquid composed of aluminium nanoparticles and ferritin proteins for printing and forming 3D shapes and structures. In total, this bioenergetic liquid exhibits increased energy output and performance, enhanced dispersion and oxidation stability, lower activation temperatures, and greater processability and functionality.

[1] Materials and Manufacturing Directorate, Air Force Research Lab, Wright-Patterson AFB, Ohio 45433, USA. [2] 711th Human Performance Wing, Air Force Research Lab, Wright-Patterson AFB, Ohio 45433, USA. Correspondence and requests for materials should be addressed to R.R.N. (email: Rajesh.naik@us.af.mil).

Energetic liquids consist of alkanes, biomass-based fuels, nitrogen-rich heterocyclic ionic liquids and combustible solvents[1–3]. Liquids offer advantages over solid powders in the form of lower activation temperatures, higher pressures and burn rates, greater volume expansion, and the ability to be printed or patterned[4]. However, liquids are traditionally limited by low energy densities ($<1\,kJ\,g^{-1}$) and slow combustion kinetics due to their high carbon, oxygen and nitrogen content[1,5]. For these reasons, they primarily function as fuels and to a much lesser extent as explosives. To circumvent these low energy densities, metal nanopowders have been added to liquids in order to supplement their overall energy output[6–9]. Metal nanomaterials possess high energy densities and are well suited to forming stable colloids and nanoparticle dispersions in many different aqueous and organic solvent systems[9–12]. For example, boron powders have been used as an additive in energetic nitrogen-rich ionic liquids and organic fuels to increase energy output[6,8], while aluminium nanoparticles (nAl) have been dispersed in organic fuels at low weight %[7,11].

The combination of nAl with strong oxidizing agents offers a reactive source of energy-releasing materials for propellants, explosives, gas generators, biocides and pyrotechnics[12–16]. However, the combustion properties are strictly dependent on processing conditions, choice of oxidizing agent, extent of particle agglomeration, the amount and type of polymeric binders used, and the physical form/material phase of the energetic material (that is, solid or liquid). To date, the properties and performance of energetic nAl materials have greatly improved through the development and discovery of new oxidizing materials and/or implementation of precise assembly and functionalization approaches directed at achieving better composites and reducing mass transport between components[17,18]. The use of biologically derived oxidizing materials in the form of iron oxyhydroxide-loaded ferritin proteins and malarial hemozoin crystals composed of polymerized haem groups, in addition to a diverse assortment of unique non-biological materials, has provided some new oxidizing agents and gains in performance[19–23]. However, current energetic nAl-based materials are manufactured as nanopowders due to severe limitations associated with producing stable dispersions in liquids. Additionally, it remains challenging to overcome the inherently low mass fractions of nAl in liquid systems because of increased viscosities and severe particle agglomeration that result from the high surface area of nAl (ref. 7). Consequently, these low mass fractions reduce the overall energy output of the energetic liquid and prevent its usefulness.

Nanoscale liquids composed of protein ionic liquids (cationized protein/anion pair) represent multifunctional solvent systems that exhibit excellent dispersion and uniformity, yield high protein content and offer the potential to disperse high mass fractions of energetic nanoparticles and overcome particle agglomeration[24–30]. Protein ionic liquids are also particularly appealing because of their unique activity and biological functionality. Protein-based ionic liquids have less than 2% water content and exhibit fluid behaviour at room temperature. They have zero vapour pressure, are stable at high temperatures to ~150 °C and are environmentally safe[25]. Protein-based ionic liquids feature all the desired properties of both organic ionic liquids and aqueous solvents, and more importantly retain their biological function/activity[24,27]. To date, ionic liquid proteins have been created using ferritin, lipase, lysozyme, myoglobin, DNA and a plant virus[26,28–31]. For energetic applications, previous work has shown that ferritin assembled with nAl led to the formation of multi-layer biothermite powders with enhanced and tunable energetic properties[19].

In this study, we utilize a ferritin protein liquid as a solvent to stabilize, disperse and load high concentrations of nAl particles in order to create energetic liquids/inks without the use of additional binders or additives. The protein liquid also serves as a stable oxidizing liquid for nAl by supplying biologically derived iron oxide ferrihydrite nanoparticles (FeO(OH)) loaded within the protein cavity of ferritin. Energetically, this bio-liquid containing nAl possesses increased energy output, enhanced performance, greater oxidation and dispersion stability, improved functionality, and expanded processability for printing and moulding energetic structures.

## Results

**Creation of ferritin ionic liquid.** Ferritin protein ionic liquids were prepared using the procedure developed by Mann et al.[28], whereby ferritin was cationized using dimethylaminopropylamine and electrostatically paired with an anionic polymer surfactant (poly(ethylene glycol) 4-nonylphenyl 3-sulfopropyl ether potassium salt ($C_9H_{19}$-$C_6H_4$-$(OCH_2CH_2)_{20}O(CH_2)_3SO_3^-$)) (ref. 29). Synthesis of ferritin-based ionic liquids was monitored by titration of cationized ferritin with increasing amounts of polymer anion. During the titration, the zeta potential of cationized ferritin increased from an initial value of $+13\,mV$ in the absence of anion to $-11\,mV$ following titration with anions (Supplementary Fig. 1). This negative value indicated the presence of excess negatively charged polymer, but, more

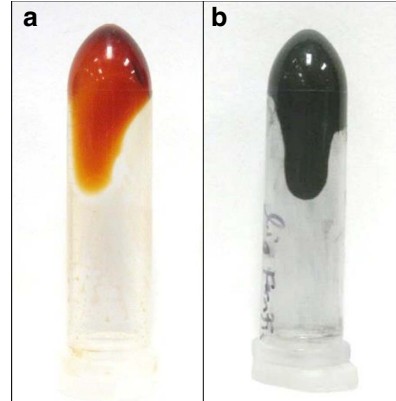
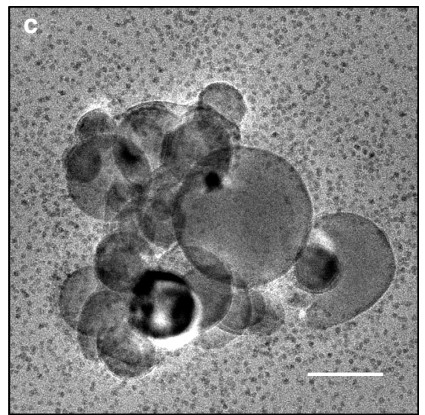

**Figure 1 | Synthesis and loading of nAl@ferritin liquid.** Digital images of (**a**) ferritin protein ionic liquid after synthesis and (**b**) ferritin protein ionic liquid loaded with nAl particles (20 wt%). (**c**) TEM micrograph of nAl dispersed in ferritin protein ionic liquid (~6 nm iron cores visible) after being reconstituted and diluted in water. Scale bar, 100 nm.

importantly, that all of the positive charges on cationized ferritin (zeta potential $+13\,mV$) were electrostatically balanced with an equal number of negative charges from the polymer anion. From the titration plot, we calculated that $\sim993$ polymer anions are needed to electrostatically balance the positively charged sites of the cationized ferritin protein and form the ionic nanoconstruct (Supplementary Fig. 1). This value is roughly threefold higher than the previously determined number of 264 polymer anions reported to electrostatically induce complexation of an equal number of positive charges (264 cationized sites) on cationized ferritin[28]. By comparison, cationization of ferritin resulted in the addition of $\sim192$ positively charged sites by MALDI mass spectrometry of light-chain ferritin (assembled ferritin cage = 90% light chain/10% heavy chain; Supplementary Fig. 2). This represented a lower number of cationized sites than previously reported (264 cationized groups) and was likely due to ineffective coupling[28]. The cationized ferritin/anion pair was further purified from excess anion via extensive dialysis and lyophilized to form a viscous orange protein ionic liquid due to the presence of the ferrihydrite core (Fig. 1a). We determined the stoichiometry of the final ionic liquid ferritin material after dialysis to contain 1,400 polymer anions per cationized ferritin protein cage at a protein concentration of $215\,mg\,ml^{-1}$ based on the absorbance of the iron oxide core of the reconstituted ferritin ionic liquid in water at 420 nm and final weight of solvent-free ferritin ionic liquid (Supplementary Figs 3,4)[32]. This is slightly higher than the number of anions determined from the titration plot of zeta potential and $\sim5.5$ times higher than previously reported[28]. This is likely due to incomplete dialysis of the polymer anion.

**Loading and dispersion of nAl in ferritin ionic liquid.** We exploited the solvent properties of ferritin protein ionic liquids by loading the liquid ferritin with a maximum of $\sim20\,wt\%$ nAl particles. Notably, this represents a higher mass loading of nAl for liquid solvents than ferrofluids ($\sim13\%$ by weight)[33]. To obtain nAl-loaded protein liquids, we reconstituted the ferritin protein ionic liquid in water to form a dilute protein solution, added $20\,wt\%$ of nAl particles dispersed in water, sonicated the suspension for 10 min (Supplementary Fig. 5) and lyophilized the nAl-loaded liquid to remove all the water over several days. After loading and lyophilization, the nAl particles were successfully dispersed in the ferritin protein liquid as demonstrated by a slightly more viscous black liquid (Fig. 1b) and contained less than $3\,wt\%$ water by thermal gravimetric analysis (TGA; Supplementary Fig. 6). By microscopy, the nAl particles appeared solvated by a matrix of cationized ferritin proteins containing spherical 6 nm iron oxide particles after reconstitution and dilution in water as determined by transmission electron microscopy (TEM (Fig. 1c), elemental dispersive X-ray spectroscopy mapping (EDAX) and dark field microscopy; Supplementary Fig. 7). Rheological studies of the material demonstrated that it behaved like a structured fluid, which initially displayed Newtonian characteristics, but upon ageing over a day at room temperature became a shear-thinning viscoelastic fluid due to the complex associative interaction between nAl and ferritin ionic liquid (Fig. 2a). The ageing process of nAl@ferritin ionic liquid resulted in shear yielding behaviour similar to ferritin ionic liquid without nAl added, but caused an overall increase in viscosity with the modulus increasing by an order of magnitude. For the experimental timescales studied, the self-organization process of transitioning from a Newtonian to viscoelastic fluid was shear- and time-dependent. However, the material always tended to structurally organize even in the nonlinear viscoelastic regime where fluidity is dominant (Fig. 2b).

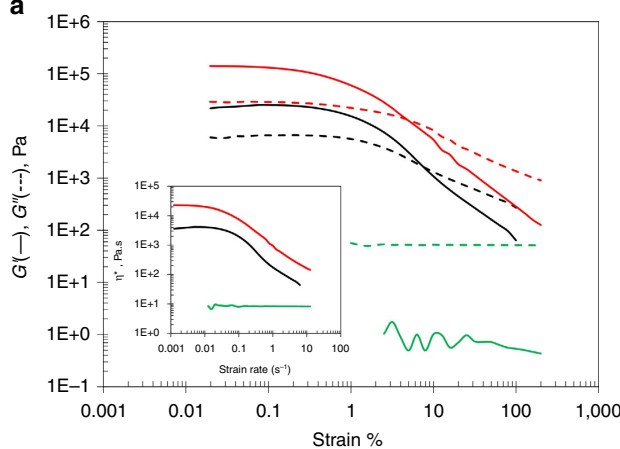

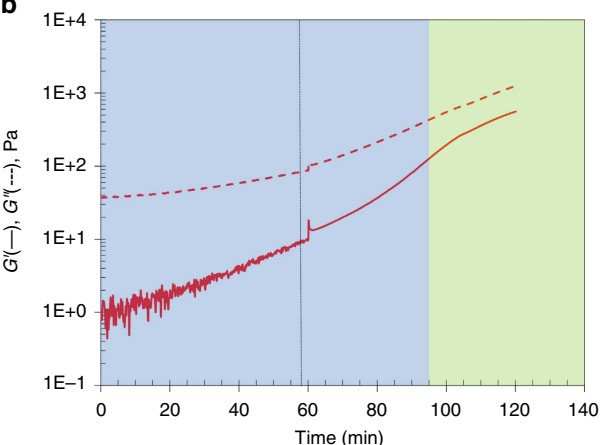

**Figure 2 | Rheological characteristics of nAl@ferritin liquids. (a)** Strain sweep mapping at an oscillatory frequency of 1 Hz. Corresponding shear flow during strain sweep (inset). Ferritin liquid only (black line), nAl@Ferritin liquid (green line), aged nAl@Ferritin liquid (red line). **(b)** Structural evolution at 10% strain and an oscillatory frequency of 1 Hz. Dashed line indicates stopping and restarting of oscillatory strain and the memory effect of restructuring. Blue region indicates linear viscoelastic regime and green region indicates nonlinear viscoelastic regime.

From the rheological tests, there was no direct evidence to suggest that microphase separation was occurring during sample ageing. The thermal dependence and occurrence of a single melt transition (Supplementary Fig. 8) indicated that nAl and ferritin ionic liquid formed an associative network, which is influenced by temperature.

The dispersion capability of ferritin ionic liquids for nAl was also measured by quartz crystal microbalance (QCM). QCM showed strong binding interactions of the ferritin ionic liquid with an aluminium oxide-coated sensor via large frequency changes and only weak interactions with the polymer anion. In contrast, the interactions between anion and aluminium oxide were easily displaced by water (Supplementary Fig. 9). These binding interactions are consistent with the high binding affinity of cationized ferritin to the negatively charged surface of aluminium oxide as previously described[19]. This suggests that upon addition of nAl to the ferritin protein ionic liquid, the cationized ferritin is attracted to the nAl surface favouring binding, while the anion is partially displaced. At the same time, polymer anions are attracted to the cationized ferritin layer on nAl, forming a second layer. This dispersion mechanism is supported by the observation of a supramolecularly layered shell

of ferritin coating the nAl surface (Fig. 1c) by TEM of reconstituted nAl@ferritin IL in water and is consistent with the dispersion mechanism of negatively charged $SiO_2$ particles in a pure organic ionic liquid[34]. In this capacity, the cationized ferritin coating on nAl prevented aggregation in the concentrated ferritin ionic liquid through steric and repulsive forces, promoted dispersion, and enabled high mass fractions. At higher loadings above 20 wt%, aluminium nanoparticles began to agglomerate and the nAl-loaded protein liquid exhibited a dry dough-like consistency.

By comparison, we surveyed the ability of aqueous solvents, organic solvents and a simple organic ionic liquid to effectively disperse nAl. As a result, common nanoparticle dispersion solvents of water, ethanol, chloroform and hexane failed to disperse nAl at low wt% and showed immediate and visible sedimentation of nanoparticles (Supplementary Fig. 10). In the case of nAl in water, nAl was rapidly oxidized to $Al(OH)_3$ after 2 days at room temperature[19]. Alternatively, simple imidazolium-based ionic liquids are well known for their exceptional solvent properties and ability to form stable nanoparticle suspensions[34]. As a result, we examined the dispersion of nAl in 1-butyl-3-methyl-imidazolium chloride ([BMIM]Cl) in order to benchmark the nAl@ferritin protein ionic liquid. Unlike the conventional solvents used above, [BMIM]Cl ionic liquid successfully produced a stable nAl dispersion (Supplementary Fig. 10). All solvents resulted in larger nAl aggregates by particle size analysis and much broader size distributions with the largest sizes occurring in ethanol (Supplementary Fig. 11). Consequently, we also tested and compared the dispersion stability of nAl in [BMIM]Cl and ferritin ionic liquid after centrifugation. Centrifugation at 14,000 r.p.m. for 1 h had little effect on the nAl@ferritin ionic liquid, but showed visible signs of nAl sedimentation in [BMIM]Cl (Supplementary Fig. 12).

**Energetic properties of bioenergetic liquid.** The energetic properties of nAl-dispersed energetic protein liquids were characterized by simultaneous TGA and differential thermal analysis (DTA) from 25 to 900 °C in a 100% argon environment to prevent air oxidation[35]. DTA analysis is routinely used to characterize energetic materials and provides details about exothermic events, total amount of energy released from materials, stoichiometry of aluminium to oxidizing agent, and how much nAl was consumed during combustion[36]. As a benchmark, the DTA profile of the ferritin protein ionic liquid revealed a small and broad endothermic region from ~60 to 390 °C due to denaturation and decomposition of the ferritin protein cage and polymer anion (Supplementary Fig. 13), a shifted exothermic peak at ~440 °C and a broad exothermic region at ~470–600 °C from the internal oxidation of protein residues, an endothermic region that rapidly decayed at ~630 °C, and an overall enthalpy of $-1.0\,kJ\,g^{-1}$ (Fig. 3a). Similarly, the thermal behaviour of ferritin ionic liquid is consistent with other proteins in inert atmospheres[37]. For nAl dispersed in an apoferritin protein ionic liquid without any oxidizing material (no FeO(OH) core inside ferritin), DTA showed a broad exotherm with two major exothermic peaks at ~440 and 580 °C due to sequential oxidation events of apoferritin and nAl as observed above for the ferritin ionic liquid alone, an endothermic region that started at ~625 °C due to aluminium melting, and a total energy output of $+2.4\,kJ\,g^{-1}$ (Fig. 3a)[38]. By comparison, nAl in the absence of protein yielded a characteristic sharp endothermic peak for nAl melting at 670 °C and a broad exotherm equal to an energy output of $+3.3\,kJ\,g^{-1}$ (Fig. 3a) due to mild oxidation. This suggests that the energetic contribution from the high content of carbon, nitrogen and oxygen (~50 wt%

of ferritin) of the apoferritin protein ionic liquid was negligible on the energy output of nAl and produced less heat. In general, this is consistent with low energy densities of organic hydrocarbon-based fuels[1,5]. Nonetheless, the presence of the high protein content during combustion aids in decreasing diffusion distance between reactive components and likely promotes increased gas production, higher pressures and increased flame speeds[8].

Thermal analysis of the nAl@ferritin ionic liquid ($2nAl + 3\,FeO(OH)_{ferritin} \rightarrow 2Al_2O_3 + 3Fe + 3OH^- + Energy$) at an equivalence ratio of 8.2 ($\phi_{eq} = (fuel/oxidizer)_{actual}/(fuel/oxidizer)_{stoichiometric}$) generated a broad exothermic reaction over the entire temperature range and yielded an energy output of $+11.3\,kJ\,g^{-1}$ (Fig. 3a). Similar to other nAl-based nanothermite materials, the reaction was self-sustaining due to low bulk density of nAl, low thermal conductivity, and close interfacial contact between nAl fuel and ferritin oxidizer[39–41]. For reaction with nAl (~40,000 atoms/80 nm nAl particle), multiple ferritin cages are needed with each ferritin protein cage providing an iron oxyhydroxide core composed of 4,500 iron atoms and 9,000 oxygen atoms[42]. For the matching and complementary nanopowder, the nAl@ferritin biothermite produced a lower energy output of $+8.9\,kJ\,g^{-1}$ at an equivalence ratio of 5.5. Also, the powder showed two endothermic peaks for aluminium melting and ferritin decomposition and resulted in incomplete combustion of the nAl fuel (Fig. 3b). Also, comparison of the energetic liquid state versus nanopowder revealed a higher energy output, more energy released at lower temperatures and complete combustion for the liquid. This suggests that combustion of the energetic liquid can be ignited at lower temperatures and that the overall diffusion of reactive components is much improved in the liquid phase. More generally, our energetic bio-liquids exhibited faster combustion kinetics and released much more energy than nAl powders assembled with periodate salts[2], ammonium perchlorate[43], graphene oxide[44], copper oxide[45], molybdenum oxide[46] or fluorinated polymers[47], but slightly less reactive than our energetic powder containing hemozion crystals[20].

Using colour high-speed digital video, we recorded the combustion of our bio-derived energetic liquid in an open air-vented fragmentation chamber. For all liquid samples, we ignited a 5 μl drop on a syringe tip and recorded the combustion event. In the absence of nAl, the ferritin protein ionic liquid exhibited slow and consistent burning that persisted for 1.65 s due to the waxy nature of the liquid (Fig. 3c). Similarly, nAl@apoferritin liquid with no iron oxide present slowly burned with minimal combustion (Supplementary Fig. 14). For nAl@ferritin liquid, the combustion images showed an initial burning of the nAl@ferritin protein liquid followed by an intense burst of energy due to combustion of nAl with the iron oxide of ferritin at ~400 ms (Fig. 3d). We wrote a few simple patterns on glass or stainless-steel fine mesh screen using nAl@ferritin liquid with a line resolution of ~2 mm (Fig. 4a). EDAX mapping of these drawn shapes showed a uniform distribution of Al and Fe over the entire inked pattern and a composition of 6 wt% Fe and 15 wt% Al (Fig. 4a). The 15 wt% Al quantitated by EDAX was slightly lower than the starting 20 wt% of nAl added during processing into the final material, although this is based on the assumption that we expected no significant mass loss during preparation of the final nAl@ferritin ionic liquid given the lack of purification steps. When ignited at a single point, the nAl@ferritin liquid ink quickly combusted and propagated along the patterned glass or mesh screen surface (Supplementary Fig. 15, Fig. 4d).

**Processing of bioenergetic liquid into moulded structures.** Additionally, we explored whether energetic protein liquids

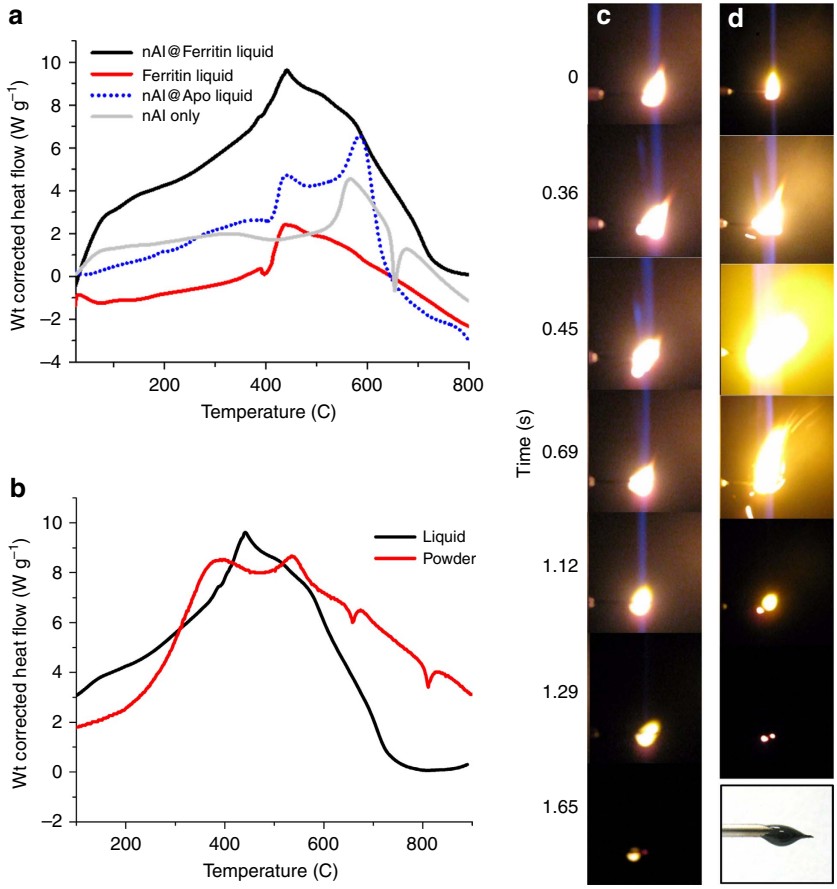

**Figure 3 | Characterization of energetic properties.** (**a**) DTA plot of nAl@ferritin liquid, ferritin liquid only and nAl@apoferritin liquid (minus iron oxide). (**b**) DTA plot of nAl@ferritin liquid compared with nAl@ferritin biothermite nanopowder. High-speed camera images showing combustion of a 5 μl drop of (**c**) ferritin only protein liquid on the tip of a syringe needle at selected time frames. (**d**) nAl@ferritin ionic liquid.

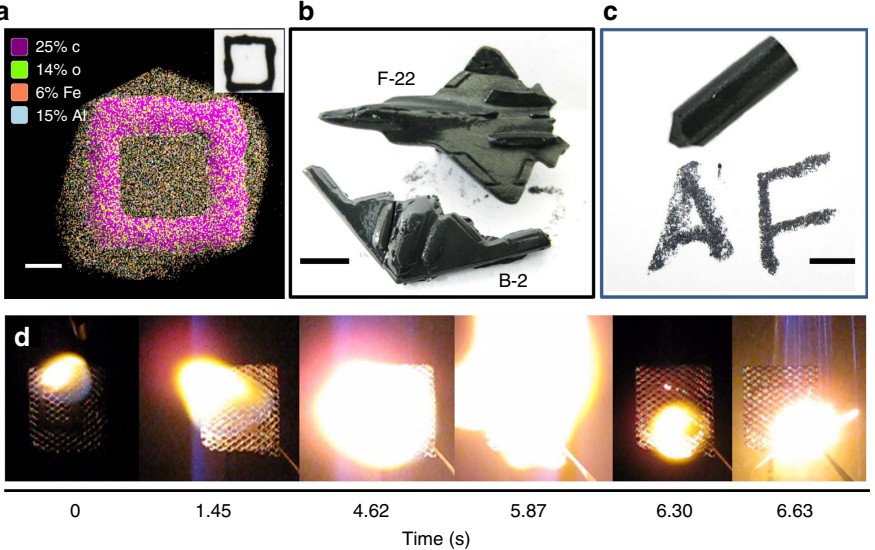

**Figure 4 | Printing and 3D moulding of energetic nAl@ferritin liquids.** (**a**) EDAX elemental map of square drawn with nAl@ferritin liquid on glass showing overlay of Al, Fe, C and O along with corresponding wt% of each. Scale bar, 1 mm. (Inset) Optical image of a square drawn on a glass slide using nAl@ferritin liquid ink. (**b**) Image of an energetic miniature 1:1,260 scale replica of F-22 fighter aircraft and B-2 bomber created by filling a PDMS mould with nAl@ferritin liquid and freeze-casting. Scale bar equals 5.5 mm. (**c**) Image of an energetic crayon created by filling a PDMS mould with nAl@ferritin liquid and freeze-casting and 'AF' handwritten on paper using the energetic crayon. Scale bar equals 7 mm. (**d**) Combustion images captured from high-speed digital video at selected time frames of a square pattern on stainless-steel fine mesh screen drawn using nAl@ferritin liquid.

containing nAl could be processed and fabricated into simple three- dimensional (3D) structures, geometries and shapes[48]. 3D printing of energetic materials offers the ability to create advanced design concepts, new complex shapes and structures, execute new operational capabilities, and exert precise spatial control of combustion for advanced munitions (that is, fragmented disassembly)[49]. As a proof of concept, we created a cylindrical crayon-like structure and a miniature 1:1,260 scale F-22 fighter aircraft or B2 bomber by freeze-casting $\sim 0.6$–$1.0\,g$ of nAl@ferritin protein liquid ink in corresponding PDMS moulds (Fig. 4b,c). The nAl@ferritin as-cast structures retained their shape after removal from the mould for several hours at room temperature, but, over extended periods of time, the 3D moulded structures slowly melted and transitioned back to their original liquid-phase form (Supplementary Fig. 16). Nonetheless, the bioenergetic liquid remained stable against oxidation over 4 weeks of ageing by X-ray photoelectron spectroscopy (XPS) analysis and showed no change in nAl particle size by particle sizing measurements on a sucrose gradient (Supplementary Fig. 17). In total, these materials represent stable liquids with excellent dispersion and oxidation stability over time. Using the energetic crayon, we were able to write on paper, glass and metal surfaces (Fig. 4). This provides an effective method to transfer and write energetic materials onto any contaminated surface and neutralize any biological/chemical agents by combustion of the ink[50]. Alternatively, we can likely prevent melting and phase change of our 3D energetic structures by tuning the physical properties of the ferritin protein ionic liquid. For example, based on similar principles of organic ionic liquids, we can decrease the length of the alkyl chain within the polymer anion to increase the melting point of the ferritin protein ionic liquid.

Ferritin protein ionic liquids served as both water-free solvents and strong oxidizing agents for dispersion and combustion of energetic nanomaterials. Also, energetic ferritin protein liquids functioned as excellent inks for writing, printing or stamping energetic materials onto different substrates and also for creation of 3D shapes and structures. To further improve the energetic properties of the nAl@ferritin ionic liquid, the iron oxide core present within the cavity of ferritin could be replaced with $MnO_2$ or CuO nanoparticles (larger $\Delta H_f$) to produce higher combustion enthalpies with nAl. These new materials enable the direct printing of precise fragmentation patterns in munitions, dialable energetic effects and decontamination applicator pens. As bioenergetic liquids, they offer many advantages over traditional nanopowders in terms of increased energy output, stability against oxidation, increased dispersion stability at high wt%, lower activation temperatures, enhanced combustion kinetics and greater functionality. Additionally, the potential for increased scalability using industrial bioreactors will extend the capabilities of these materials. Given the diverse collection of potential new protein ionic liquids with different biological activities and oxidizing capabilities, new formulations composed of multi-functional liquids can be created to tailor the energetic properties of nAl on demand with longer shelf-lives.

## Methods

**Synthesis of ferritin protein ionic liquid.** Horse spleen ferritin (Sigma) or apoferritin was cationized by adding 1 ml of ferritin stock solution, 1 ml of neat 3-dimethylaminopropylamine and 5 mg of EDC (1-ethyl-3-(3-dimethylamino-propyl)carbodiimide hydrochloride) to 30 ml of a 0.1 M phosphate buffer pH 7. Ferritin was cationized for 3 h and then dialysed in water using dialysis tubing with MWCO of $7,000\,g\,mol^{-1}$ over 2 days with three water changes. Two millilitres of anionic polymer surfactant poly(ethylene glycol) 4-nonylphenyl 3-sulfopropyl ether was first diluted in 15 ml of double deionized water at 50 °C to promote dissolution of viscous polymer. The diluted polymer anion was slowly added directly to the dialysed cationized ferritin to form a cation/anion pair. The cationized ferritin/polymer anion mixture was further dialysed over 2 days with three water changes to remove excess unpaired polymer anion. After dialysis, the mixture was lyophilized for 2 days to remove water. The lyophilized powder was slightly heated to $\sim 55$ °C for 20 min to form the ferritin protein ionic liquid.

**Synthesis of energetic nAl@ferritin protein ionic liquid.** Aluminium nano-particles (80 nm, 80% active Al content) passivated with an amorphous aluminium oxide were obtained from Novacentrix, Inc. Five milligrams of aluminium nano-particle powder was suspended in 5 ml of deionized water and sonicated using an ultrasonic bath for 5 min to disperse nAl particles. This was immediately added to $\sim 20\,mg$ of ferritin protein ionic liquid reconstituted in 30 ml of water and sonicated for 5 min. The nAl@ferritin dispersion was frozen using liquid nitrogen and lyophilized for 2 days to remove water. After 2 days, the lyophilized powder was slightly heated to $\sim 55$ °C for 20 min to form the energetic nAl@ferritin protein ionic liquid. Similarly, we also dispersed nAl in apoferritin ionic liquid as described above.

**Characterization.** Bright-field TEM images were obtained on a Phillips CM200 transmission electron microscope operating at 200 kV. TEM samples were pre-pared by reconstituting $\sim 10\,\mu l$ of nAl@ferritin ionic liquid in 500 µl of water and pipetting 10 µl of this sample onto a 3 mm 200 mesh copper TEM grid coated with ultrathin carbon film (Ted Pella) and air dried. For EDAX maps, an FEI $C_s$-corrected Titan TEM microscope operating at 300 kV was used. SEM images and EDAX maps were obtained on a FEI Quanta SEM microscope operating at 10 kV and a working distance of 25 mm. A square or star pattern was drawn on a Si wafer using nAl@ferritin ionic liquid and mounted on a standard SEM puck. Zeta potential measurements were obtained on a Malvern nano series Zetasizer using a disposable folded capillary cell (Malvern, DTS1070). Rheological information was obtained using an ARES-G2 rheometer equipped with a force rebalance transducer operating at an oscillatory frequency of 1 Hz using 25 mm stainless-steel parallel plates with environmental conditions set to 25 °C and $N_2$ atmosphere. QCM measurements were obtained on a Q-Sense E4 quartz crystal microbalance with dissipation (QCM-D) system with flow modules. Aluminium oxide ($\alpha$-$Al_2O_3$) coated QCM sensors (Q-Sense, QSX-309) were cleaned by UV/ozone treatment for 10 min, immersed in a 2% SDS solution for 30 min, washed thoroughly with water and dried with $N_2$ gas. Ferritin ionic liquid or polymer anion at $100\,\mu g\,ml^{-1}$ reconstituted in water was flowed across QCM sensors at a flow rate of $0.17\,ml\,min^{-1}$ and monitored vs time by measuring the third overtone frequency at a constant temperature of 23 °C. High-resolution XPS spectra were collected on an M-Probe Surface Science XPS spectrometer. Spectra were collected at a spot size of 800 µm, 0.01 eV step and averaged over 100 scans.

**Energetic characterization.** Simultaneous thermogravimetric analysis (TGA) and DTA measurements were performed in a TA Instruments SDT Q 600. Samples ($\sim 5\,mg$) were placed into a tarred alumina crucible with an empty alumina crucible serving as the reference. All data were collected in dynamic mode under flowing argon ($100\,ml\,min^{-1}$) from room temperature up to 800 °C at a rate of $20\,°C\,min^{-1}$. Heats of combustion in $kJ\,g^{-1}$ were obtained by measuring the area of the DTA peaks plotted vs time (s) from 25 to 900 °C. Combustion experiments were performed by placing $\sim 5\,\mu l$ of the respective nAl@ferritin liquid or ferritin liquid only onto the tip of a syringe needle in a vented fragmentation chamber under an air atmosphere. The liquid drops were initiated by a flame lighter. A NAC Image Technology Memrecam GX8 digital high-speed video camera, collecting full-frame, full-colour images at 5,000 frames s⁻¹, was used to record the com-bustion event. PDMS moulds were prepared by thoroughly mixing and degassing 1 g of catalyst added to 9 g of precursor. After mixing, PDS moulds were cured at $\sim 40$ °C for 18 h. nAl@ferritin ionic liquids were cast in moulds and frozen at $-20$ C for 10 min. The moulded nAl@ferritin ionic liquid structures were carefully removed from moulds.

**Data availability.** The data that support the findings of this study are available from the corresponding author upon request.

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

## Acknowledgements

This work was funded by Air Force Office of Scientific Research.

## Author contributions

J.M.S. performed the synthesis, processing and characterization of materials. R.M. performed rheological experiments on materials. P.B.D. and R.R.N. conceived ideas and designed the experiments. All authors contributed to data analysis, discussions and manuscript writing.

## Additional information

**Competing interests:** The authors declare no competing financial interests.

