## [Peer Review File · Nature Communications]

Reviewers' comments:

Reviewer #1 (Remarks to the Author):

The manuscript entitled "Creation of Energetic Biothermite Inks Using Ferritin Liquid Protein" describes the dispersion of aluminum nanoparticles in a solvent-free ferritin liquid at a high mass percent to produce a solvent-free liquid thermite. The concept is original and the experimental approach is reasonably sound, but there are significant gaps in the analyses of this new material and the precursors. Moreover, there is a distinct lack of detail in a number of areas including:

1. Figure captions
2. Intellectual/scientific descriptions of some of the experimental data
3. Effective referencing of the SI figures

With that in mind, the results are exciting and I believe that if the manuscript is significantly improved, it could be accepted in Nature communications. This would, however, require major corrections.

Comments:

1. Line 72.

The authors describe the ultra-stability of protein ionic liquids. What does this refer to? There have been a number of papers showing that solvent-free liquid proteins unfold at 200°C, e.g., see papers by Brogan et al..

2. Line 76.

"To date, liquid proteins have been created using ferritin, lipase, myoglobin, DNA, and a plant virus." Lysozyme has also been successfully used to produce a liquid protein (see Journal Of Physical Chemistry B, Volume: 117, Issue: 28, Pages: 8400-8407)

3. Line 77.

"For energetic applications, previous work showed that ferritin assembled with nAl led to the formation of multi-layer biothermite powders with enhanced and tunable energetic properties." There is no reference. This should be added.

4. Line 86.

The Authors describe the cationization of ferritin, but do not characterize the cationized protein. Cationization efficiencies are highly dependent on the pH, the presence of a buffer, type of buffer, the number of equivalents etc. Mass spectrometry of the cationized protein would provide this information.

5. Line 92.

As mentioned above, please refer specifically to figures in the SI. In the case of Fig. S1, expressing the zeta potential as a function of charge stoichiometry (site:polymer chain) would be more informative.

6. Line 96.

The authors state: "From the titration plot, we calculated ~993 polymer anions per cationized ferritin protein are needed to neutralize all the positive charges." Why is this the case? Was the system at equilibrium? This needs to be explained.

7. Line 99.

The authors state: "The cationized ferritin/anion pair was further purified from excess anion via extensive dialysis..." How can one be sure that there is no excess surfactant, especially in light of the vast excess used in the first instance? The Authors should calculate the final charge:surfactant stoichiometry. They could weight a dry sample, re-dissolve and measure the absorbance at 280. They could then use a bicinchoninic acid (BCA assay) to determine the protein concentration, and the residual absorbance would be coming from the nonylphenyl moiety of the surfactant. The surfactant extinction could then be used to calculate the protein:surfactant stoichiometry.

8. Line 107.

The authors state: "To obtain nAl loaded protein liquids, we reconstituted the ferritin protein ionic liquid in water to form a dilute protein solution, added 20 wt% of nAl particles dispersed in water, sonicated suspension for 10 minutes, and lyophilized the nAl loaded liquid to remove all the water over several days." How can they be sure that the protein cages are still intact after sonication? TEM with negative stain would help to show this.

9. Line 112

The authors state: "Rheology of the nAl@ferritin liquid showed less viscoelastic behavior and increased shear thinning at higher frequencies as compared to ferritin liquid with no nAl." Where is the data? This need to be present.

10. Line 117 "...elemental dispersive X-ray spectroscopy mapping (EDAX), and dark field microscopy (Figure 1, Supplemental Material)". The particles do not look dispersed in Figure 1c. Moreover, there need to be more detail when referring to Figures in either the body or SI.

11. Line 126

The high loading capacity of the ferritin melt is rationalized by interactions between the cationized ferritin and the nAl particles, resulting in displacement of the surfactant. If this was the case, one would expect microphase separation, with polymer- and protein-nAl rich regions. This needs to be discussed further or supported by another technique (SAXS?)

12. Line 140

The authors describe the use of apoferritin, but the synthesis of the apo is not mentioned, nor the source.

13. Line 141

The authors state: "...showed a broad exotherm with 2 major peaks at ~ 410°C and 600°C, an endothermic region that started at ~625°C due to aluminum melting, and a total energy output of +2.4 kJ/g (Figure 2A)" What is the feature at 410°C? Protein decomposition?

14. Line 189

The authors state: "For example, we can replace the currently used weakly interacting polymer anion with a stronger coordinating anion to increase the melting point of the ferritin protein liquid above room temperature." Could the authors expand on this? There is no detail and changing the headgroup may result in the loss of the liquid phase completely.

15. Line 200

The authors state: "To further improve the properties of the nAl@ferritin the iron oxide present within the cavity of ferritin could be replaced with a stronger oxidizing agent." Could the authors expand on this? Please provide an example.

Reviewer #2 (Remarks to the Author):

In this manuscript, the authors present a new type of energetic mixture produced by the mixing of Al Nps with ferritin protein ionic liquid.

They characterized the heat of reaction at 11kJ/g which makes this material interesting in the overall community.

It seems that the combustion is sustained (which is not clear to me as the data are provided).

The results obtained here are very interesting and could promote new insights, directions and interest in the field of energetic material. But, in its state, some parts of the manuscript present overstatements, mostly due to lacking details and quantitative data in support. The manuscript should be revised to avoid the journal readers from lack of understanding and confusion.

The idea to use biologically derived oxidizing materials with Al has already been explored and therefore does not constitute a transformative novelty. In the previous work, the ferritin protein ionic liquid was used as solvent to stabilize and load a high concentration of Al Nps. The increment of this paper is that the protein liquid also serves both as a water-free solvent and as an oxidizing agent since ferrihydrite iron particles are loaded in the protein cavities.

As a consequence, this paper should describe the reaction pathway and give a better contribution of each ingredient (ferrihydrite iron and protein molecule) in the combustion process which is lacking.

Which reactions are responsible for the heat release in Al+ferritin protein ? The DSC curves merit to be more quantitatively analyzed.

What is the role of C, N in the reaction. Is the oxygen content in the amino-acid sequences of importance (ratio to iron hydroxide?)

How is calculated the optimized ratio between Al and ferritin protein ionic liquid ?

How many Fe and O atoms per protein ?

L25-26, the authors mentioned that they could create an ultra-stable and highly energetic liquid ◊ this is false since later in the paper, we see that the composition is not stable at room temperature after a few hours.

L66-67 : « represent new multifunctional solvent systems that exhibit excellent dispersion and uniformity, yield high protein content... »

This has to be supported by data. Characterization of Al nanopowder into the protein solvent is required and compared with what was obtained in other solvents. Same comment on the uniformity.

L77-79 : Previous work showed that ferritin assembled with nAl led to the formation of multi-layer biothermite powders with enhanced and tunable energetic properties. At least the reference is required.

L131. Stability over time has to be addressed in a more convincing manner.

Fig2. Caption must use the same terminology as in the text. The reference curve which is nAl in apoferritin protein is not mentioned as it is in the graph ! Is it the blue curve ? If yes, I do not see the exotherm with the two major peaks.

Red curve : I don't see the slightly shifted endotherm region at 600°C.

What is the range of temperature integration to find 11.3 kJ/g ?

L155 : what do the authors mean when using « more reactive » ? Does it refer to the combustion front propagation ? or to the capability to be ignited faster ?

L164-170 : is the combustion sustained ?

L173 : How was calculated the percentage of Fe and Al in the final material ?

Can the authors comment on the scalability of this new material as well as on the applicative interest since at room temperature, the solvent melts.

Response to Reviewers:

Firstly, we thank the reviewers for their thoroughful suggestion and constructive critique of the manuscript. Below is a response to the points raised by the reviewers.

Reviewer #1

1. Figure captions

– *We have add more details to the figure captions.*

2. Intellectual/scientific descriptions of some of the experimental data

– *Include in text and materials and methods (see below)*

3. Effective referencing of the SI figures

– *We have referenced the SI figure appropriately in the text.*

Comments:

1. Line 72.

The authors describe the ultra-stability of protein ionic liquids. What does this refer to? There have been a number of papers showing that solvent-free liquid proteins unfold at 200°C, e.g., see papers by Brogan et al..

- *We agree with the referee and corrected the statement to read "... are stable at high temperatures to ~150C. Also, we added the appropriate reference describing the general temperature stability (Reference #27).*

2. Line 76.

"To date, liquid proteins have been created using ferritin, lipase, myoglobin, DNA, and a plant virus." Lysozyme has also been successfully used to produce a liquid protein (see Journal Of Physical Chemistry B, Volume: 117, Issue: 28, Pages: 8400-8407)

- *We included lysozyme in the list of protein liquids and added respective reference (Reference #26).*

3. Line 77.

"For energetic applications, previous work showed that ferritin assembled with nAl led to the formation of multi-layer biothermite powders with enhanced and tunable energetic properties." There is no reference. This should be added.

- *We added the appropriate reference (Reference #8).*

4. Line 86.

The Authors describe the cationization of ferritin, but do not characterize the cationized protein. Cationization efficiencies are highly dependent on the pH, the presence of a buffer, type of buffer, the number of equivalents etc. Mass spectrometry of the cationized protein would provide this information.

- *We performed mass spec on native and cationized ferritin and included the mass spectra of both as a new supplemental figure (Figure S2). Also, we calculated the number of cationized sites based on mass spec of the ferritin light chain and provide additional discussion on pg 5.*

5. Line 92.

As mentioned above, please refer specifically to figures in the SI. In the case of Fig. S1, expressing the zeta potential as a function of charge stoichiometry (site:polymer chain) would be more informative.

- *We agree with referee that the current plot is not clear. We replotted zeta potential vs # of charges per cationized site and replaced as a new Supplemental figure (Figure S1).*

6. Line 96.

The authors state: "From the titration plot, we calculated ~993 polymer anions per cationized ferritin protein are needed to neutralize all the positive charges." Why is this the case? Was the system at equilibrium? This needs to be explained.

- *We based our explanation and synthesis according to Mann et al. whereby an equal number of polymer anions are needed to electrostatically induce complexation with cationized sites on ferritin to produce the ionic nanoconstruct.*

7. Line 99.

The authors state: "The cationized ferritin/anion pair was further purified from excess anion via extensive dialysis..." How can one be sure that there is no excess surfactant, especially in light of the vast excess used in the first instance? The Authors should calculate the final charge:surfactant stoichiometry. They could weight a dry sample, re-dissolve and measure the absorbance at 280. They could then use a bicinchoninic acid (BCA assay) to determine the protein concentration, and the residual absorbance would be coming from the nonylphenyl moiety of the surfactant. The surfactant extinction could then be used to calculate the protein:surfactant stoichiometry.

- *We used the absorbance of the iron oxide core at 420 nm to determine the concentration of ferritin in reconstituted ionic liquid based on the reference (Mat et al.: Archives of Biochem Biophys 1978, 190:720-725). The converted mass of ferritin was then subtracted from the total final mass of dry ferritin ionic liquid to determined mass of anion present and then ultimately to determine a stoichiometry. In total we determined there are about 1452 polymer anions per ferritin cage. We included the standard concentration plot of ferritin and corresponding absorbance spectra at different concentrations.*

8. Line 107.

The authors state: "To obtain nAl loaded protein liquids, we reconstituted the ferritin protein ionic liquid in water to form a dilute protein solution, added 20 wt% of nAl particles dispersed in water, sonicated suspension for 10 minutes, and lyophilized the nAl loaded liquid to remove all the water over several days." How can they be sure that the protein cages are still intact after sonication? TEM with negative stain would help to show this.

- *We sonicated ferritin for 1 hour, negatively stained sample, and obtained a TEM image showing negatively stained protein cages. The TEM image of ferritin is included in supplemental material as Figure S5 and shows intact protein cages surrounding the iron oxide core.*

9. Line 112

The authors state: "Rheology of the nAl@ferritin liquid showed less viscoelastic behavior and increased shear thinning at higher frequencies as compared to ferritin liquid with no nAl." Where is the data? This needs to be present.

- *We added the missing data in Figure 2 in the manuscript and included a more detailed analysis of rheological behavior in text.*

10. Line 117 "...elemental dispersive X-ray spectroscopy mapping (EDAX), and dark field microscopy (Figure 1, Supplemental Material)". The particles do not look dispersed in Figure 1c. Moreover, there needs to be more detail when referring to Figures in either the body or SI.

- *We agree with the referee regarding the dispersion of nAl particles in the TEM image and clarified our statement. The corresponding TEM image shows the nAl@ferritin ionic liquid after reconstitution in water. Consequently, this image nor preparation of the TEM sample necessarily reflect an accurate depiction of how well the particles are actually dispersed in the true solvent-free ferritin ionic liquid.*

11. Line 126

The high loading capacity of the ferritin melt is rationalized by interactions between the cationized ferritin and the nAl particles, resulting in displacement of the surfactant. If this was the case, one would expect microphase separation, with polymer- and protein-nAl rich regions. This needs to be discussed further or supported by another technique (SAXS?)

- *We revised discussion based on a similar dispersion mechanism of NPS in organic ionic liquids. Using rheology, we showed there was no direct evidence to suggest that microphase separation was occurring.*

12. Line 140

The authors describe the use of apoferritin, but the synthesis of the apo is not mentioned, nor the source.

- *We included a short description for the preparation of apoferritin ionic liquid in experimental section. Apoferritin is obtained from commercial source and modified similar to ferritin.*

13. Line 141

The authors state: " ...showed a broad exotherm with 2 major peaks at ~ 410°C and 600°C, an endothermic region that started at ~625°C due to aluminum melting, and a total energy output of +2.4 kJ/g (Figure 2A)" What is the feature at 410°C? Protein decomposition?

- *The exothermic peak ~440C is consistent with oxidation of protein residues and similar to the thermal behavior of other proteins, whereas the endothermic peaks before 410 are due to protein denaturation and decomposition. We added this description and more detail about thermal behavior in text.*

14. Line 189

The authors state: " For example, we can replace the currently used weakly interacting polymer anion with a stronger coordinating anion to increase the melting point of the ferritin protein liquid above room temperature." Could the authors expand on this? There is no detail and changing the headgroup may result in the loss of the liquid phase completely.

- *We revised our statement to include an alternate way to modify melting point based on similar principles of organic ionic liquids.*

15. Line 200

The authors state: " To further improve the properties of the nAl@ferritin the iron oxide present within the cavity of ferritin could be replaced with a stronger oxidizing agent." Could the authors expand on this? Please provide an example.

- *We added an example and provided a brief explanation on pg 13.*

Reviewer #2:

In this manuscript, the authors present a new type of energetic mixture produced by the mixing of Al Nps with ferritin protein ionic liquid.

They characterized the heat of reaction at 11kJ/g which makes this material interesting in the overall community. It seems that the combustion is sustained (which is not clear to me as the data are provided). The results obtained here are very interesting and could promote new insights, directions and interest in the field of energetic material. But, in its state, some parts of the manuscript present overstatements, mostly due to lacking details and quantitative data in support. The manuscript should be revised to avoid the journal readers from lack of understanding and confusion.

The idea to use biologically derived oxidizing materials with Al has already been explored and therefore does not constitute a transformative novelty. In the previous work, the ferritin protein ionic liquid was used as solvent to stabilize and load a high concentration of Al Nps. The increment

of this paper is that the protein liquid also serves both as a water-free solvent and as an oxidizing agent since ferrihydrite iron particles are loaded in the protein cavities.

As a consequence, this paper should describe the reaction pathway and give a better contribution of each ingredient (ferrihydrite iron and protein molecule) in the combustion process which is lacking. Which reactions are responsible for the heat release in Al+ferritin protein ? The DSC curves merit to be more quantitatively analyzed.

- *We included more detail on DTA curves in terms of individual components and explained contributions of protein and iron oxide core on the amount of energy released.*

What is the role of C, N in the reaction. Is the oxygen content in the amino-acid sequences of importance (ratio to iron hydroxide?) How is calculated the optimized ratio between Al and ferritin protein ionic liquid ?

- *We described in more detail the energetic contribution of carbon, nitrogen, and oxygen from protein on nAl by comparing DTA curves of nAl with and without protein (no iron oxide). We included the equivalence ratio of iron oxide to nAl and the corresponding equation in the main text.*

How many Fe and O atoms per protein ?

- *We included the total numbers of Fe and O atoms per iron oxide core of ferritin in the discussion on pg 10.*

125-26, the authors mentioned that they could create an ultra-stable and highly energetic liquid this is false since later in the paper, we see that the composition is not stable at room temperature after a few hours.

- *We agree with referee that the liquid materials do not necessarily possess great phase stability in the solid form as 3D molded structures, however, they represent stable "liquids" with excellent dispersion and oxidation stability over time. We included more detail and new supplemental figures on pg 12 and Figure S17.*

166-67 : « represent new multifunctional solvent systems that exhibit excellent dispersion and uniformity, yield high protein content.... » This has to be supported by data. Characterization of Al nanopowder into the protein solvent is required and compared with what was obtained in other solvents. Same comment on the uniformity.

- *We included an image and size data on dispersion of nAl in aqueous solvents, organic solvents, and a traditional organic ionic liquid in supplemental material (Figure S10 and Figure S11). Also, we included protein concentration to show high protein content. To show uniformity of nAl within protein liquid, we used EDAX mapping and dark field microscopy and included images in Figure 4 and supplemental material (Figure S7).*

L77-79 : Previous work showed that ferritin assembled with nAl led to the formation of multi-layer biothermite powders with enhanced and tunable energetic properties. At least the reference is required.

- *We added appropriate reference (Reference #8).*

L131. Stability over time has to be addressed in a more convincing manner.

- *We included a size plot and XPS analysis of initial sample and after 4 weeks of ageing to show no change in NP size or oxidation state of aluminum (Figure S17).*

Fig2. Caption must use the same terminology as in the text. The reference curve which is nAl in apoferritin protein is not mentioned as it is in the graph ! Is it the blue curve ? If yes, I do not see the exotherm with the two major peaks. Red curve : I don't see the slightly shifted endotherm region at 600°C.

- *We corrected the figure plots to include nAl in apoferritin liquid and revised discussion.*

What is the range of temperature integration to find 11.3 kJ/g ?

- *Heats of combustion in kJ/g was obtained by integrating the area of the DTA peaks plotted vs time (sec) from 25°C - 900°C at 20°C/min. We included this description in text.*

L155 : what do the authors mean when using « more reactive » ? Does it refer to the combustion front propagation ? or to the capability to be ignited faster ?

- *We clarified statement describing “more reactive” to mean faster combustion kinetics in text.*

L164-170 : is the combustion sustained ? Explain and compare to other nanothermites published.

L173 : How was calculated the percentage of Fe and Al in the final material ?

- *We included additional references on nanothermites for comparison and included detail on sustained combustion of our material. We calculated the wt% of nAl in ferritin ionic liquid based on the initial mass of nAl added and initial mass of ferritin ionic liquid (no water) used before reconstitution, sonication, and lyophilization. We expect no mass loss of nAl or ferritin ionic liquid during processing into the final material since there are no dialysis or purification steps.*

Can the authors comment on the scalability of this new material as well as on the applicative interest since at room temperature, the solvent melts.

- *We included some detail on feasibility of material scalability and added a few applications on pg 13.*

REVIEWERS' COMMENTS:

Reviewer #1 (Remarks to the Author):

The authors have addressed my concerns and the manuscript is much improved.

Reviewer #2 (Remarks to the Author):

Authors made good corrections. Manuscript can be publish as is.